# Sleep/Wake Behavior and EEG Signatures of the TgF344-AD Rat Model at the Prodromal Stage

**DOI:** 10.3390/ijms21239290

**Published:** 2020-12-05

**Authors:** Matthias Kreuzer, Glenda L. Keating, Thomas Fenzl, Lorenz Härtner, Christopher G. Sinon, Ihab Hajjar, Vincent Ciavatta, David B. Rye, Paul S. García

**Affiliations:** 1Department of Anesthesiology and Intensive Care, School of Medicine, Technical University of Munich, 81675 Munich, Germany; m.kreuzer@tum.de (M.K.); thomas.Fenzl@tum.de (T.F.); 2Department of Anesthesiology, Emory University, Atlanta, GA 30322, USA; gkeatin@emory.edu (G.L.K.); christopher.george.sinon@emory.edu (C.G.S.); vciavat@emory.edu (V.C.); 3Department of Neurology, Emory University School of Medicine, Atlanta, GA 30322, USA; rlsrye@gmail.com; 4Department of Neurology, Division of Neurobiology, Medical University Innsbruck, 6020 Innsbruck, Austria; lorenz.haertner@web.de; 5Neuroscience Graduate Program, Emory University, Atlanta, GA 30322, USA; 6Division of Geriatrics, Department of Medicine, Emory University School of Medicine, Atlanta, GA 30322, USA; ihabhajjar@emory.edu; 7Department of Anesthesiology, Neuroanesthesia Division, Columbia University Medical Center, New York Presbyterian Hospital, New York, NY 11355, USA

**Keywords:** Alzheimer’s, electroencephalography, sleep architecture

## Abstract

Transgenic modification of the two most common genes (APPsw, PS1ΔE9) related to familial Alzheimer’s disease (AD) in rats has produced a rodent model that develops pathognomonic signs of AD without genetic tau-protein modification. We used 17-month-old AD rats (*n* = 8) and age-matched controls (AC, *n* = 7) to evaluate differences in sleep behavior and EEG features during wakefulness (WAKE), non-rapid eye movement sleep (NREM), and rapid eye movement sleep (REM) over 24-h EEG recording (12:12h dark–light cycle). We discovered that AD rats had more sleep–wake transitions and an increased probability of shorter REM and NREM bouts. AD rats also expressed a more uniform distribution of the relative spectral power. Through analysis of information content in the EEG using entropy of difference, AD animals demonstrated less EEG information during WAKE, but more information during NREM. This seems to indicate a limited range of changes in EEG activity that could be caused by an AD-induced change in inhibitory network function as reflected by increased GABAAR-β2 expression but no increase in GAD-67 in AD animals. In conclusion, this transgenic rat model of Alzheimer’s disease demonstrates less obvious EEG features of WAKE during wakefulness and less canonical features of sleep during sleep.

## 1. Introduction

With an aging society, more people will suffer from neurodegenerative disorders like Alzheimer’s disease (AD) [1] and present an increasing economic burden to relatives, societies, and healthcare providers [2]. Currently, no effective prevention or treatment strategies for AD exist. This necessitates the use of realistic animal models to fully understand the development and progression of AD. The transgenic rat model (TgF344-AD) [3] includes the two most common genes associated with familial AD: the mutant human amyloid precursor protein (APPsw) and presenilin 1 (PS1ΔE9) and demonstrates neuropathological findings pathognomonic of neurodegeneration not seen in all AD models. This model offers many advantages over murine models of AD. It is the first rodent model to display a full pathology consistent with human AD through an acceleration of age-dependent accrual of the β-amyloid protein (Aβ) in the presence of wild-type tau and all within the animal’s normal lifespan: dense and diffuse amyloid plaques, high levels of soluble Aβ oligomers, cerebral amyloid angiopathy, hyperphosphorylated tau and paired helical filaments, cerebral vacuoles (Hirano bodies, granulovacuolar), robust neuroinflammation, cognitive impairment, and substantial cortical neuronal loss [3]. In this model, neuropathology and memory deficits that are consistent with AD typically develop after 15 months of age. In this study, we focused on the sleep architecture and electroencephalographic (EEG) characteristics of different vigilance states of the TgF344-AD rat model at an early disease stage and compared the results to age-matched healthy control animals. We relate these changes with biochemical evidence of changes in the inhibitory network function. Sleep disturbances are common in patients with neurodegenerative diseases [4].

Alzheimer’s disease is associated with fragmentation of sleep [4] and characteristic changes of the EEG [5]. Results from experiments conducted in mice suggest a relationship between changes to the sleep–wake cycle and pathogenesis and progression of Alzheimer’s disease [6]. Hence, an investigation of sleep and EEG characteristics at early stages of AD may help to develop early detection procedures for AD as well as provide insight into disease progression. In general, EEG amplitudes decrease with age during wakefulness [7,8], sleep [9], as well as general anesthesia [10,11,12]. Cortical thinning presents one cause for this age/power relationship [13,14]. This feature seems exaggerated in AD patients [15]. Hence, the thinner cortex present in severe AD is associated with a decrease in EEG power in these patients [15]. In rats, age-related decreases in power also occur [16]. Despite these similarities, humans with AD do not simply express accelerated or worsening age-related EEG changes. Alterations of the human EEG specific for AD are associated with stages of AD progression. As mild symptoms of AD develop, beta (14–40 Hz) power decreases. As AD progresses to a more severe dementia, EEG delta (0.5–4 Hz) power increases and alpha (8–14 Hz) power decreases [5]. Because of the heterogeneity associated with sporadic AD in humans, less information is available regarding EEG characteristics at presymptomatic stages or at an early stage of AD, often termed mild cognitive impairment (MCI), where patients begin to express subtle signs of memory loss. Recent results suggest an increased relative theta power in humans that are amyloid-beta positive, but show no symptoms of AD [17]. Early disruption of signaling in inhibitory networks appears to play a central role in the neurophysiologic changes observed in pre- and early-disease states [18,19]. To investigate brain dynamics at early stages, the use of animal models known to develop AD confers large advantages.

## 2. Results

### 2.1. Demographics

We did not observe significant differences in weight, age, and gender between the AC (*n* = 7) and AD (*n* = 8) animals used for the sleep and EEG analyses. The AC animals weighed 422 (93) g (median and median absolute deviation (mad)) and the AD animals weighed 455 (29) g. The AUC analysis revealed a non-significant effect of 0.61 (95% CI: 0.30–0.89). The AC animals were 17 (1) months old, the AD ones—17 (0) months old with an AUC of 0.53 (0.25–0.79) indicating no effect. The chi-squared test indicated no significant differences in gender distribution (AC: 3 females, 4 males; AD: 2 females, 6 males; *p* = 0.906).

The animals (*n* = 4) used for the protein expression results were 17–21 months old in the AC group (median: 18 months). The median age for the AD group used for biochemical analysis was 20 months (range: 17–23 months, *n* = 4). These mild age differences were not considered statistically significant (AUC: 0.70 (0.37–0.96) for the AC groups and 0.69 (0.31–1) for the AD groups).

### 2.2. Sleep Macroarchitecture Based on the 10 s Sleep Scoring

Figure 1 shows the hypnogram for each animal included in the study over the entire 24 h observation period. Figure 2 describes the proportion of vigilance states throughout the 24 h observation period in 2 h bins. Total time spent in the vigilance states (WAKE, NREM, REM) was not significantly different between the groups.

### 2.3. Bout Length Based on the 10 s Sleep Scoring

We found differences in the bout length distribution for all the vigilance states in the inactive period as well as during NREM in the active period. The cumulative probability plots from Figure 3 show that the AD animals expressed a steeper slope at shorter bouts during NREM indicating a higher probability of shorter bouts than the AC animals. This observation also holds true for REM.

### 2.4. Transition between WAKE and Sleep Levels

All results from the quantitative EEG analysis are based on the rescoring of vigilance states using a distinct 2 h recording period scored in 4 s EEG episodes. The AUC for the SLEEP/WAKE transitions was AUC = 0.73 (0.46–0.94) for the inactive phase and AUC = 0.78 (0.5–1) for the active phase indicating an acceptable or fair effect of strain on the transition behavior with more transitions in the AD groups. The AD animals seemed to have an increased number of transitions. Figure 4 presents the dot plots for the transitions between WAKE and sleep periods for the 12 h observation period as well as the 2 h period used for the EEG analysis. The AD animals in the active period are less stable in the WAKE state and less stable in overall sleep during the inactive period.

### 2.5. AD Rats Express Different Sleep Microarchitecture as Evaluated by Spectral EEG Properties

During the 2 h period for quantitative EEG analysis, we did not have REM episodes in one AC rat and three AD rats during the lights-off phase. During the lights-on phase, we also did not find REM episodes in three AD rats. For one AD rat during the lights-off phase and two AD rats during the lights-on phase, we did not find REM episodes at all. Two intracranial extradural recording electrodes were used for quantitative EEG (qEEG) analysis: a caudal electrode (see Methods) positioned over the neocortex and a rostral electrode that records hippocampal information. Although we did not find significant differences in the absolute spectral power among the groups, we found increased relative spectral power in the higher frequencies in the AD rats (Figure 5 and Appendix A). We detected stronger effects in the rostral lead. In the context of classical EEG frequency bands, we determined that EEG power in the delta band range is lower in the AD animals, especially during WAKE and NREM. That explains the increased spectral power in the higher frequency bands. Power in the theta band range as well as in the beta band range was increased during WAKE, NREM, sleep, and REM sleep in the active phase. EEG power in the alpha band range was increased in the AD animals during inactive WAKE and active and inactive NREM sleep. For the caudal electrode, the relative power of the AD animals showed similar effects (see Appendix A). Appendix A contains the corresponding AUC values for the analysis of EEG power in the frequency band ranges together with the 95% confidence intervals.

### 2.6. AD Rats Express Different Sleep Microarchitecture as Evaluated by Entropic EEG Properties

In order to evaluate the significance of the increase in relative power in the higher frequency bands for AD animals, we examined the complexity of the qEEG information. Therefore, we used the entropy of difference (EoD) that is based on evaluating the sign of the difference between neighboring values. The primary finding is a significant change in the EoD with the vigilance state, i.e., between WAKE and NREM sleep in the AC animals that is absent in the AD animals. Figure 6 shows the corresponding details for the EoD (τ = 1) of the caudal EEG and Appendix A—for the EoD (τ = 1) for the rostral EEG. Further, when comparing the AC and AD animals at respective vigilance states, the EoD (τ = 1) of the AD group is significantly (rostral: 0.78 [0.45–1]; caudal: 0.96 [0.81–1]) lower, indicating a reduced complexity in the WAKE EEG that may correspond with decreased information content. During NREM sleep, the information content as evaluated by EoD (τ = 1) was significantly (e.g., AUC = 0.12 [0–0.35] for the rostral EEG during the active phase) higher in the AD rats (Table 1). During REM sleep, we only observed effects in the caudal EEG with a significantly higher EoD (AUC = 0.86 [0.57–1]; caudal, active phase). The detailed results are presented in Appendix A (rostral) and Appendix A (caudal). Appendix A contains detailed statistical results.

### 2.7. Protein Expression

We did not find significant differences (AUC: 0.60, 95% CI: 0.2 to 1) in the GAD-67 expression between the AC and AD animals. However, we found a perfect separation in the GABA_A_R β_2_ expression (AUC = 1) between AC and AD indicating a very strong effect. Figure 7 presents the single analysis results.

## 3. Discussion

We present a first characterization of sleeping behavior and EEG activity in the TgF344-AD rat model at an early disease stage and a comparison to AC rats. We were able to show AD-related altered sleep behavior and EEG characteristics. In short, these changes in the sleep macro-architecture point towards a more fragmented sleep behavior in the prodromal AD animals. The changes in the sleep microarchitecture (EEG) indicate a reduction in the change of EEG dynamics between WAKE and NREM sleep in the AD animals.

### 3.1. Early-Stage AD Influences Sleep Macro-Architecture

The vigilance states’ proportions were not significantly different between AD and AC rats over the 24 h observation period. Still, there seems to be a difference in the REM proportions between AD and AC animals. AD animals tend to have less REM sleep. These findings are in line with results from recent studies. In experiments with J20 mice, these animals showed reduced REM sleep if compared to controls [20]. Further, an observed reduction in REM sleep in patients is associated with a higher all-cause dementia risk [21]. Our results point towards that direction as well. AD animals showed a different distribution of WAKE, NREM, or REM bout length during the inactive period and had a higher probability of only short WAKE and REM bouts and “medium length” NREM bouts. This finding is supported by the observation of an increased number of sleep/wake transitions in the AD rats. Of course, a direct translation to humans is hard. AD patients seem to wake up more often during nighttime, take longer to go back to sleep [22], and seem to have a lower sleep efficiency [23]. The increased WAKE amount, the decreased REM amount, and the higher number of transitions further develop as AD progresses [24,25]. The fragmentation may be explained by the finding that negative impact on the sleep/wake cycle increases cerebral Aβ levels [26], i.e., AD and associated sleep disturbances trigger a downward spiral. Even at a preclinical stage, individuals with amyloid deposition (Aβ42 levels) showed lower sleep quality, but not quantity [27].

All results of the quantitative EEG analysis were obtained from a distinct 2 h recording period (refer to Methods for details), scored in 4 s EEG episodes rather than 10 s for a higher resolution. Epoch lengths of 4 s or 10 s are broadly used in rodent sleep research, although when dealing with the analysis of stage transitions, for example, a 4 s epoch has explicitly been suggested [28]. This is in line with our experience in sleep scoring applying 4 s epochs to score fast and abrupt transitions [29,30].

### 3.2. Differences in Sleep Microarchitecture between AD and AC Rats

Besides differences in sleep macro-architecture, we found the sleep microarchitecture, i.e., the EEG features, to be significantly different between AD and AC animals. Our spectral analysis using relative EEG power revealed significant differences in the composition of the EEG between AC and AD animals. The EEG power was more uniformly distributed in the AD animals than in the AC group. This is reflected by the significant increase in EEG power in the higher frequencies, especially in the rostral EEG during WAKE and NREM sleep. The strain also had an effect on relative power in the low frequencies corresponding to the EEG delta range. AD rats seemed to have lower relative power in these low frequencies as indicated by AUC > 0.7. Similar results were described for other rodent models for AD. J20, AβPPswe/PS1ΔE9, and APP23 amyloidosis mice also feature lower (relatively) power in the low frequencies and increased (relatively) power in the high frequencies [20,31,32,33].

To correctly compare and interpret our findings from our studies using animals that show first signs of behavioral/cognitive impairments, they rather have to be matched with findings with patients expressing a less severe phenotype (i.e., MCI) instead of those who have progressed to AD. EEG characteristics of MCI patients and of patients with frank AD differ. MCI patients demonstrate increased EEG theta [34] and alpha together with a decreased delta power [35]. Further, NREM sleep’s slow wave activity (0.8–4.6 Hz) decreases with age and correlates with reduced memory performance [36]. Presenile AD patients also have lower relative delta power [37]. So our findings seem in line with these observations. In general, early AD EEG properties seem to stand in strong contrast to EEG characteristics of patients with progressed Alzheimer’s. These patients show an EEG oscillation shift towards slower frequencies with increased delta and theta EEG spectral power and decreased alpha and beta power [5].

It should be noted that rodents have, by comparison to humans, a thinner neocortex relative to the brain size. Therefore, oscillations from deeper structures make up a larger proportion of the overall EEG power with cranial electrodes. This is especially relevant for theta power, best seen over the hippocampal formation. In rodents, theta power in our rostral electrode is strongly indicative of the WAKE and REM sleep states (see Figure 5). Corticocortical communication was the focus in our examination of complexity (see Figure 6). For these results, we elected to emphasize the caudal lead (over the prefrontal cortex), as changes in the information content in the rostral lead might be assumed to contain more information from the hippocampus. Further examinations reveal that relationships between sleep macro-architecture and the EEG properties of the sleep microarchitecture seem to exist. Our AD animals showed increased sleep fragmentation. In humans, patients with primary or subjective insomnia express higher relative beta power [38,39]. One possible interpretation is that NREM sleep is less restful for AD rats, because the cortex is activated to a greater degree (Table 1). Our observations of increased EEG complexity using EoD during REM and NREM sleep in the fast dynamics and the increased number of sleep/wake transitions during the inactive period support this theory. Recent research revealed a potential link between the power spectral centroid and the first derivative of the signal, here, the EEG [40]. During WAKE, AD rats had lower EoD, indicative of lower signal complexity or information content in the EEG. As for the spectral parameters, translation to human patients may be difficult. In a MEG study, (early-stage) AD patients had lower signal complexity as measured by approximate entropy and Lempel–Ziv complexity compared to healthy controls during the awake, eyes closed states [41]. In another study, early AD patients showed higher permutation entropy in parietal and occipital EEG [42]. This seemingly opposite finding may be explained by the fact that the recordings were at relaxed eyes closed conditions and alpha power in these regions is decreased in AD patients [43]. The wake versus sleep comparisons revealed that the AC rats’ EoD differs significantly between WAKE and NREM sleep, which is not the case in the AD group. This may indicate that the brain of the AC rats is capable of expressing a stronger change in EEG dynamics between WAKE and NREM sleep. This *dynamic range* is smaller in the AD rats, possibly implying that AD rats are “not as awake” and/or “not as well asleep” as the AC rats.

In general, we found numerous markers corresponding with sleep and EEG characteristics in patients with early AD. AD is associated with degeneration of neurotransmitter pathways involved in sleep/wake cycles. Progressive loss of basal forebrain cholinergic neurons innervating the hippocampus and cortex occurs during AD [44]. Previous lesion studies in rats demonstrated changes in high-frequency EEG activity during REM after destruction of basal forebrain-to-cortex projections [45]. Further, glutamate neurotransmission seems dysregulated by Aβ in AD [46]. Post-mortem AD human brains show evidence of impairments in glutamatergic signaling [47,48,49]. Aβ infusion into rodent brains has been associated with glutamate release and reuptake dysfunctions resulting in an excess of extracellular glutamate [50,51]. The increase in synaptic glutamate may cause aberrant glutamate receptor activation causing excitotoxic damage or death in these neurons.

In contrast, GABAergic inhibitory networks seem to remain fairly well intact during AD progression [52,53,54]. In the wake of AD-related dysregulation, over-excitation, and eventual neuronal death apparent in excitatory networks, survival of GABAergic networks alone may cause problematic imbalances in the cortical excitatory/inhibitory tone as AD progresses. Demonstrations of seizure activity in AD in human patients [55,56] and in AD mouse models [57] support this idea. However, some studies have indicated inhibitory networks’ changes via alterations in the GABA_A_R subunit expression that may be disease-related [52,53] or indicate compensatory changes in the wake of the disease as reviewed in [58].

It is known that this rodent model exhibits signs of increased cortical excitability. Interestingly, treatment with a drug that increases hippocampal signaling (cholinergic) appears to mitigate this seizure-like activity [59]. The failure of the rats in our AD group to stabilize their quiescent activity (sleeping) without intrusion of the awake phase supports the notion that suppression of neuronal activity is inadequately achieved in this disease model while quiescent. Similarly, the intrusion of sleep during waking behaviors suggests instability among vigilance states rather than a simple bias towards cortical excitation. Given the previously published presence of Aβ and neurofibrillary tangles at 16 months in a TgF344-AD rat [3], we analyzed protein expression related to GABA transmission in order to probe for possible evidence of inhibitory networks’ alterations in the early disease state. Our molecular data suggest that maintaining appropriate cortical excitation in each vigilance state is complex in these animals. Although it appears that both AC and AD groups seem to make a similar amount of inhibitory neurotransmitter at the synapses (see Figure 7), the production of GABA_A_ receptors appears to be upregulated in the disease model (increase in expression). These molecular results can only be considered preliminary evidence that changes in the synaptic balance might underlie the physiologic changes and several caveats must be considered.

First, molecular data in this study were only recorded in a small number of animals. The protein expression analysis revealed significantly increased GABA_A_R β_2_ expression in the AD animals but no significant change in GAD-67, i.e., in inhibitory synaptic transmission. Because two outliers may cause the observation of no significant result, we can draw no definitive conclusion regarding a possible effect. This increased protein production may not translate to increased functional GABA_A_ receptors on the neurons’ surface, likely resulting in altered inhibitory network function. However, we are aware of our limited sample size and much more research focused on disease-related changes in excitatory and inhibitory networks in this model is needed.

Second, although no exact age range has been established that marks the end of “early” Alzheimer’s disease in this model [3], it is important to consider that our molecular data come from animals older than our physiologic data. Therefore, these data may represent a greater progression of disease severity. Lastly, we did not quantify all the potential proteins that could contribute to excitatory and inhibitory balance. Future studies focused on membrane expression, the influence of glutamatergic, cholinergic signaling and the proportion of extrasynaptic to intrasynaptic GABA_A_ receptors will be necessary to form a more complete picture. Although the β_2_ subunit of the GABA_A_ receptor might be considered a marker of general GABA_A_ receptor levels [60], it is possible that expression of other GABA- and non-GABA-related proteins is responsible for these effects.

It remains to be empirically established whether our findings translate to the clinical setting. TgF344-AD rats contain all six tau isoforms, whereas mice only express three of the human tau isoforms [61]. A comparison of our findings from AD rats to established transgenic AD mouse models may also be difficult because of differences in disease progression. Five-months-old mice already show memory impairment [62]. For example, we did not observe a reduction in NREM sleep in our AD animals compared to the aged controls, which is in line with findings from human studies for patients with MCI. When we compare our findings to sleep and EEG of transgenic AD mouse models, we share the observation of increased hippocampal theta power. They further have increased wakefulness/decreased NREM sleep in their AβPP/PSEN1 [63] or PLB1_Triple_ group [62], a change we did not observe. This may be caused by different disease stages, or those mice had increased delta power in the 5-months-old PLB1_Triple_ mice [62].

Of course, there are some limitations in our study. We only investigated differences between AC and AD animals for one specific age range. This range corresponds to a phase the AD animals express early cognitive decline, arguably corresponding to an MCI stage of human AD. Hence, our findings may not help to design a presymptomatic biomarker for EEG-based AD detection, but is more relevant to the prodromal AD stage. At a later disease stage, the found differences between the groups may change, because AD progression strongly influences EEG features [5]. Our EEG recording setup includes a 1 Hz high-pass filter. Hence, we were not able to evaluate possible changes in the very slow frequencies corresponding to the low delta band of the EEG. This additional information would have helped us study EEG changes and sleep quality during NREM sleep in the very low frequencies in more detail. Further, our EEG was low-pass filtered at 30 Hz at recording. That is why we were not able to evaluate EEG differences at higher frequencies, something that has to be looked at in the future.

In conclusion, our study provides a characterization of sleep and EEG changes associated with the TgF344-AD model, expressing human-like AD changes during early-stage AD. Our findings are translatable, since they are consistent with observations from AD patients. Hence, our findings underline this model’s potential to investigate Alzheimer’s disease and its sleep-related manifestations.

## 4. Materials and Methods

All protocols were reviewed and approved by the Institutional Animal Care and Use Committee (IACUC) at Emory University (Atlanta, GA, USA). Animals were sourced from an internal colony of transgenic AD rats that included littermate aged controls (Emory University, Atlanta, GA, USA) [3]. The animals recovered from EEG implantation surgery (see EEG Surgery) for 7–10 days before sleep scoring commenced. We included a total of 15 animals in the investigation of sleep behavior: 7 animals in the aged control (AC) group and 8 animals in the AD group. Post-mortem tissue was collected from nine additional animals (*n* (AC) = 5, *n* (AD) = 4) for biochemical analysis. All animals used in this study were at least 17 months old and considered exhibiting frank disease in the early stage. Throughout the protocol, animals were housed under a 12:12 h light/dark cycle with food and water provided ad libitum.

### 4.1. EEG Surgery

For the purpose of EEG recording, we implanted EEG and EMG electrodes in the animals as described in [64,65]. For the implantation procedure, we anesthetized the animals with isoflurane (induction: 4–5% at 1 l/min for 2 min; maintenance: 1–3% at 1 l/min) and placed them into a stereotaxic frame. We drilled four holes in the skull and inserted two pairs of sterile 0–80 × 3/32 screw electrodes (Plastics One, Roanoke, VA, USA). We placed the first electrode pair at coordinates anteroposterior (AP): −1.5 mm, mediolateral (ML): 3.0 mm and AP: −6.3 mm, ML: 3.5 mm referenced to bregma and the second pair contralaterally at AP: +2.5 mm, ML: 1.5 mm and AP: −3.6 mm, ML: 1.5 mm. Hence, we were able to derive a caudal EEG recording that contains information on prefrontal cortical activity and a rostral EEG recording that contains information regarding hippocampal activity. Because the results of the sleep microarchitecture for the rostral and caudal EEG were quite similar, we decided to only present the results from the rostral analysis in the manuscript and present the results from the caudal EEG in the Appendix A. For EMG recording, we inserted a fine wire (40 gauge Cooner Wire, Chatsworth, CA, USA) into the left and right nuchal muscle. We covered the micro-connector (Continental Connector Corp., Middlebury, CT, USA) to which the screws and the wire were attached to with a dental acrylic compound (Plastics One, Roanoke, VA, USA) following electrode placement. We sutured the free ends caused by incision after the acrylic skullcap had dried. This left the micro-connector pins exposed and unencumbered for recurrent recording. After electrode placement, we immediately tethered the rats with a cable to a swivel commutator in a recording chamber to capture recovery EEG and sleep/wake for 48 h. We then removed the rats from their tether and the recording chamber and returned them to their home cage for 7–10 days. Following this home cage recovery time, rats were re-tethered and the baseline sleep/wake data used for our analysis were obtained. After this recovery period, we collected the data for our analyses.

### 4.2. Data Collection—EEG and EMG

We used Grass model 12A5 amplifiers for amplification, followed by EEG band pass filtering to 1–30 Hz and EMG band pass filtering to 10–40 Hz. The signals were digitized with an NI analog-to-digital converter (PCI-MIO-16E-4, National Instruments, Austin, TX, USA) at a sampling rate of 200 Hz.

### 4.3. EEG Recording Procedure

In order to record the EEG over a 24 h sleep/wake cycle, we placed the animals into a sound-attenuated, ventilated, and light-regulated environmental cubicle (89 cm high, 86 cm wide, and 74 cm deep; BRS/LVE, Davis, MD, USA), and attached them to a counterbalanced and suspended commutator via their head connectors. Each subunit within the cubicle was self-contained with individual air inlet ports, exhaust fans, and adjustable light source and with temperature controlled in the range between 24.4 °C and 26.7 °C [65]. We maintained a 12:12 h light/dark cycle with lights on (i.e., start of the inactive period) at 7 am. The rats had access to water and food ad libitum. After we tethered the animals, we started the continuous recording of EEG/EMG. Figure 8 presents exemplary EEG episodes as used in our analyses.

### 4.4. Assessment of Vigilance States

We used the Somnologica Science (Embla, Buffalo, NY, USA) software to score vigilance states in 10 s blocks. We used the scores from this analysis to evaluate the overall sleep architecture of the animals from both groups over the entire sleep/wake cycle.

In general, one scorer blind to the experimental condition evaluated the sleep/wake architecture data on an epoch-by-epoch basis. Each 10 s epoch was scored as either wakefulness (WAKE), non-rapid eye movement sleep (NREM), or rapid eye movement sleep (REM) according to the standard criteria. Animal behavior and neurophysiology may dynamically fluctuate throughout both the active (lights off) and inactive (lights on) states. To control for these dynamic changes, we limited the high-resolution quantitative EEG analysis to 2 distinct 2 h periods in each light cycle, i.e., from 2–4 am (active cycle) and 2–4 pm (inactive cycle). A 4 s resolution allows for a more detailed extraction of distinct vigilance states, i.e., the chances of being able to extract EEG episodes that contain only one state without any state transition increase. We used a MATLAB (R2015a)-based, semi-automated sleep scoring routine [30] to score selected segments with a finer time resolution of 4 s blocks. The scoring is based on spectral EEG band power parameters as described in [30,66,67]. We used these sleep classifications (with 4 s epochs) to analyze the EEG characteristics in the AC and AD animals.

### 4.5. Analysis of Sleep Architecture

For the analysis of the sleep architecture, we evaluated three different parameters: (i) the proportion of the vigilance states: WAKE, NREM sleep, and REM sleep; (ii) the bout length for each vigilance state, i.e., the time the animal resided in one state; (iii) the number of transitions between WAKE and SLEEP (NREM sleep and REM sleep combined).

While the proportion of vigilance states gives an overall impression of the amount of WAKE, NREM sleep, and REM sleep throughout the 24 h observation period, the bout length and SLEEP/WAKE transitions provide information regarding the sleep fragmentation. This seems important to our investigation, because (early) AD or MCI lead to sleep fragmentation [33,68]. Bout length and number of transitions were derived for the active and inactive periods.

### 4.6. Quantitative EEG Analysis of WAKE, NREM Sleep, and REM Sleep

We pooled the EEG of the 2 h episodes (ZT: 7–9 h, 19–21 h) to groups representing the three vigilance states. In case, we could not find a REM episode within the 2 h period, we would include the closest REM episode in the same light/dark cycle. We calculated the power spectral density (PSD) for the EEG of each vigilance state with the MATLAB pwelch function with NFFT set to 256 leading to a frequency resolution of 200 Hz/256 = 0.78 Hz. For the normalization of absolute power to relative power, we divided the absolute power by the sum of absolute power between 1 and 30 Hz.

In order to quantify the information content in the EEG at the different vigilance states, we calculated the entropy of difference (EoD) [69], a parametric measure based on Shannon entropy and closely related to the permutation entropy (PeEn) [70]. The idea of these measures is to reduce the time series (the EEG) to a series of rank (PeEn) or sign patterns (EoD). The more uniform the distribution of these patterns, the higher the entropy in the signal. In contrast, if the time series only contains one rank or sign pattern, as it is the case in the strictly monotonically increasing signal, the entropy is minimal. The nonparametric permutation entropy proved to present a robust measure of different vigilance levels during general anesthesia [71,72]. EoD presents a similar measure that quantifies the probability distribution of the sign patterns derived from the difference in amplitude between m consecutive (defined by lag parameter τ) data points. A time series of amplitude values (5, 4, 6, 2, 3, 1) would lead to following motifs (and sign patterns) of length m = 3 and τ = 1: (5,4,6)[−,+], (4,6,2)[+,−], (6,2,3)[−,+], and (2,6,1)[+,−]; for τ = 2: (5,6,3)[+,−] and (4,2,1)[[–,–]. We set the embedding dimension m of EoD to 5 and the time lag τ to 1 in order to evaluate the fast dynamics in the EEG and τ = 6 to evaluate slow dynamics. Hence, the term “fast dynamics” relates to the analysis of the information content of the fast oscillations contained in the EEG and the term slow dynamics relates to the information in the slow EEG oscillations.

### 4.7. SDS-PAGE/Western Blotting

We homogenized frozen samples of brain tissue samples derived from AD (*n* = 4) and AC (*n* = 5) in ice-cold buffer (10 mM Tris–HCl and 1 mM ethylenediaminetetraacetate (EDTA)) and centrifuged the homogenized tissue at 4 °C with 13,000× *g* for 20 min. We determined the total protein concentration of supernatants before diluting them with a sample buffer. Then, we loaded a total of 50 µg protein per lane onto SDSPAGE Criterion precast 10% gels (Bio-Rad Laboratories Inc., Hercules, CA, USA). Following electrophoresis, we transferred the protein samples to a polyvinylidene fluoride (PVDF) membrane using a semidry system (Bio-Rad Laboratories Inc.). We blocked membranes for one hour in 5% non-fat dry milk in a mixture of tris-buffered saline and Tween- 20 (TBST) (25 mM Tris, 140 mM NaCl, 3 mM KCl, and 0.1% Tween-20) and probed overnight at 4 °C with the primary antibody diluted in 5% bovine serum albumin (Sigma-Aldrich, St. Louis, MO, USA) in TBST. Then, we washed the membranes and probed for one hour with a horseradish peroxidase-conjugated secondary antibody diluted in 5% milk TBST. We performed detection of immunoreactive bands with chemiluminescence using an ECL substrate (ECL Prime, Abersham, or West Pico, Pierce) on a Chemidoc imager (Bio-Rad Laboratories Inc., Hercules, CA, USA) and quantified the bands in ImageJ (NIH, Bethesda, MD, USA) using the mean intensity. We normalized GABA_A_ receptor (GABA_A_R) subunit intensity to β-actin. The primary antibodies we used were as follows: GABA_A_R β_2_-subunit (Abcam, Cambridge, UK), Ab1560000; GAD-67 (Sigma-Aldrich, St. Louis, MO, USA), SAB4300642; β-actin (Sigma-Aldrich, St. Louis, MO, USA), A5441; the secondary antibodies were as follows: mouse NA931, rabbit NA934 (both Amersham, GE, USA; Pittsburg, PA, USA). We included the analyses of the GAD-67 and GABA_A_R β_2_-subunit expression in this manuscript. The GAD-67 expression gives information regarding the amount of GABA produced at presynaptic terminals and the GABA_A_R β_2_-subunit expression gives information regarding the amount of GABA_A_R.

### 4.8. Statistical Analysis

We performed a number of statistical tests to investigate possible differences in sleep architecture, EEG characteristics, and protein levels. We calculated the time spent in each vigilance state (WAKE, NREM, and REM) in 2 h time periods across the 24 h recording. AD rats were compared to age-matched controls using 2-way ANOVA with factors “aged control / disease” and “time” with the MATLAB’s anovan function. We used the Dunn–Sidak correction for post-hoc analysis.

For the assessment of differences of bout length between the groups, we applied the concept of a cumulative probability plot [73] in combination with a two sample Anderson–Darling test performed in MATLAB to accept or reject the null hypothesis that bout lengths of each group have the same cumulative probability distribution as already used in [74]. The cumulative probability plot enables reading the probability of bouts that are below or above a certain threshold length and present a visualization of the bout lengths’ relative distribution. We statistically compared the number of transitions using AUC and bootstrapped 95% confidence intervals (CI) using the MATLAB-based measures of effect size (MES) toolbox [75]. If the 95% CI does not include 0.5, the result is significant (*p* < 0.05). A rough classification of the effect size can be made according to the traditional academic point system: AUC values ≥ 0.9 can be interpreted as excellent; AUC ≥ 0.8 indicates a good effect; and AUC ≥ 0.7 indicates a fair effect. Lower values, 0.7 > AUC ≥ 0.6, indicate a poor effect and even lower values mean that there is no effect. In clinical applications, AUC of 0.7–0.8 is acceptable, 0.8–0.9— excellent, and 0.9–1— outstanding discrimination [76]. The AUC can have values below 0.5. In that case, the effect size can be derived using the following formula, 1–AUC, i.e., an AUC of 0.15 indicates a good effect, because 1 − 0.15 = 0.85. Further, the use and measures of effect size with 95% confidence intervals help to quantify possible differences between groups [77].

For the analysis of the EEG spectral power, we also calculated the AUC for each frequency bin and only report results as being significant if the 95% confidence interval excludes 0.5 for at least two neighboring frequencies, a procedure similar to, e.g., the one presented in [12]. We used AUC and 95% CI to evaluate differences in EoD (i) between AC and AD at different vigilance states and (ii) between WAKE and NREM for AD and AC. The latter may help to quantify the change in information content between wakefulness and sleep and hence help to evaluate the dynamic range of EEG activity. In order to evaluate differences between the protein expression in the AD and AC group, we calculated the AUC and 95% CI as well.

## Figures and Tables

**Figure 1 ijms-21-09290-f001:**
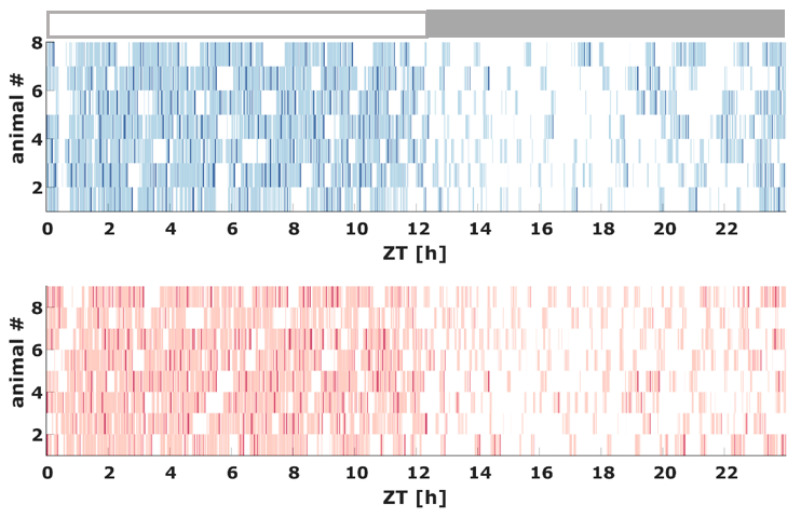
Hypnogram of the AC (top) and AD (bottom) animals over the entire 24 h observation period derived from the 10 s scoring with lights turned off after the 12th hour. White indicates WAKE, light blue (AC) or light red (AD) indicates NREM, and blue (AC) or red (AB) indicates REM. The x-axis is Zeitgeber time (ZT) with ZT = 0 indicating the start of the inactive “lights-on” phase at 7 am. The grey rectangle indicates the active “lights-off” phase. #: number

**Figure 2 ijms-21-09290-f002:**
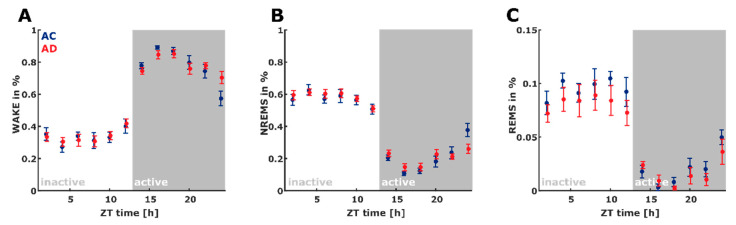
There were no significant differences in (**A**) WAKE (F_WAKE_ = 0.03, *p* = 0.857), (**B**) NREM sleep (F_NREM_ = 0.41, *p* = 0.522), and (**C**) REM sleep (F_REM_ = 3.00, *p* = 0.085) between the control (AC, blue, *n* = 7) and Alzheimer’s (AD, red, *n* = 8) animals. The distribution plots were derived from the 10 s scoring. The x-axis is Zeitgeber time (ZT) with ZT = 0 indicating the start of the inactive “lights-on” phase at 7 am. NREMS: NREM sleep; REMS: REM Sleep.

**Figure 3 ijms-21-09290-f003:**
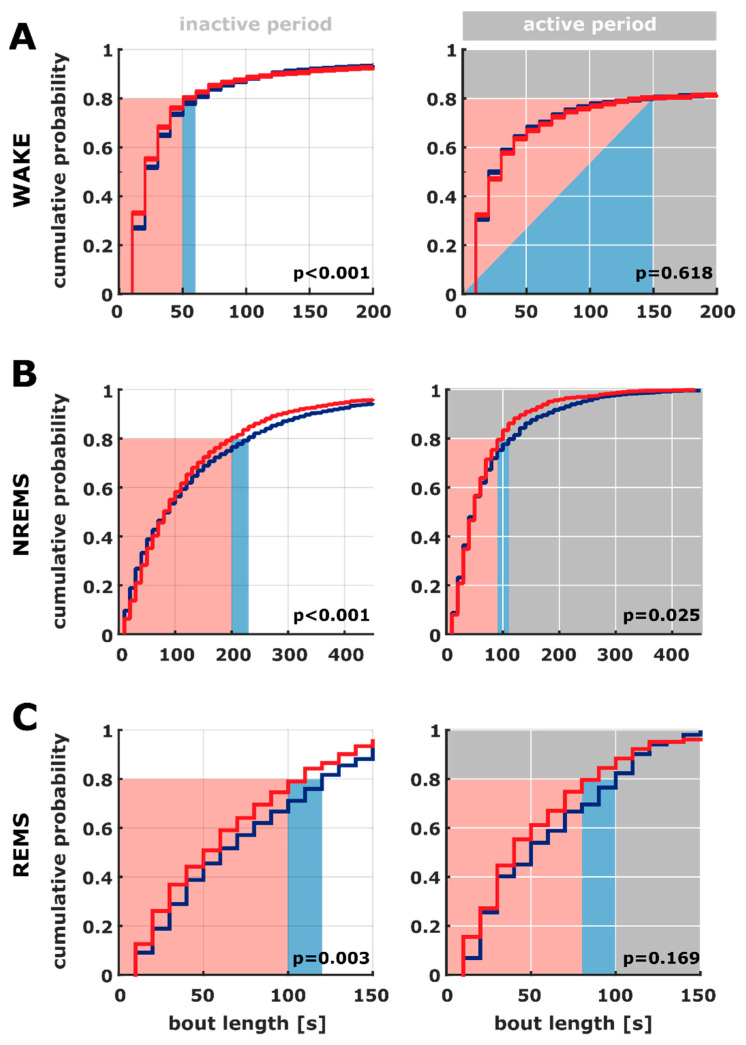
Cumulative probability plots of the different vigilance states derived from the 10 s scoring in logarithmic presentation of the inactive (left) and active (right) periods for (**A**) WAKE, (**B**) NREM sleep (NREMS), and (**C**) REM sleep (REMS). Blue: AC group, red: AD group. The bout lengths of WAKE (inactive) and NREM sleep (active and inactive) are significantly different among the groups. Except for the WAKE bouts in the active phase, the bout length distributions are significantly different. In general, the AC graphs are approaching cumulative probability 1 more gradually indicating a more uniform distribution of bout lengths than in the AD rats. The red and blue areas indicate the bout duration corridor; where 80% of the bouts were less than that time epoch.

**Figure 4 ijms-21-09290-f004:**
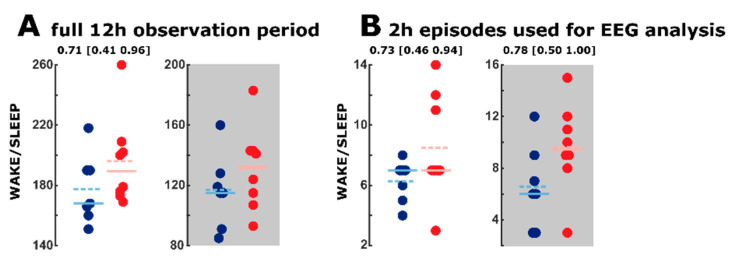
Dot plots inclusive mean (dashed line) and median (solid line) of transitions between WAKE and SLEEP of the AC group (*n* = 7, blue) and the AD (*n* = 8, red) group. (**A**) The plot represents the transition for the entire 12 h of the active or inactive period, scored in 10 s epochs. For the inactive period, we derived AUC values reflecting a fair effect. For the WAKE to SLEEP transition, AUC was 0.71 (95% CI: 0.41–0.96). (**B**) For the 2 h episodes (scored in 4 s epochs) used for quantitative EEG analysis, the AD animals showed a higher number of transitions as well.

**Figure 5 ijms-21-09290-f005:**
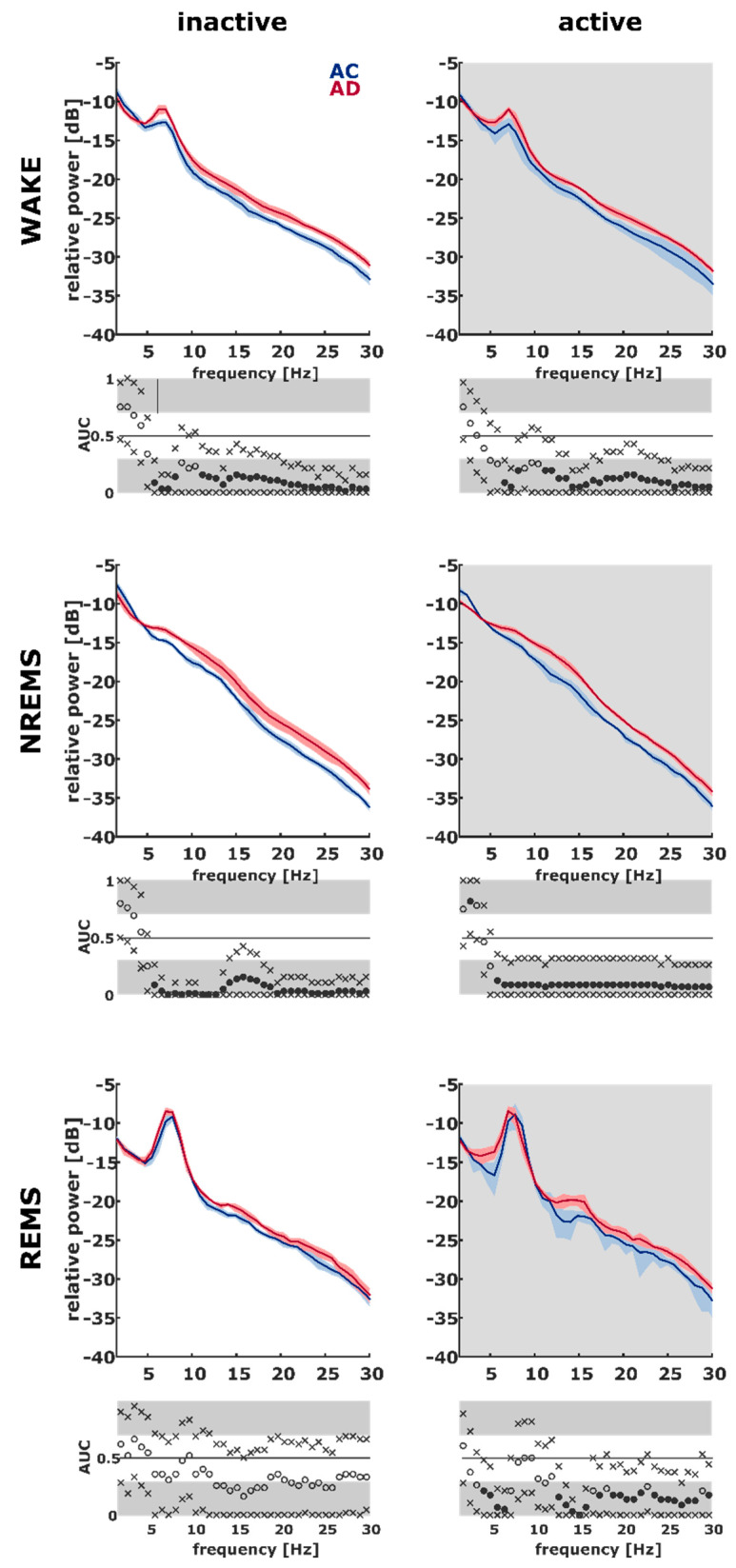
Plots of the median of the relative spectral power for the control group (AC, *n* = 7, for REM: *n* = 5, blue) and the Alzheimer’s (AD, *n* = 8, for REM: *n* = 6, red) group as recorded from the rostral electrode. The light blue and red shaded areas indicate the median absolute deviation. The accompanying AUC plots including the 95% confidence intervals represent the results of the statistical analysis. Dots indicate the calculated AUC, x signs indicate the limits of the 95% confidence intervals. A filled dot indicates a significant difference as determined by the 95% confidence interval exclusive 0.5. The areas in gray indicate an AUC > 0.7, i.e., an AUC value corresponding to an at least acceptable effect. REMS: REM sleep; NREMS: NREM sleep

**Figure 6 ijms-21-09290-f006:**
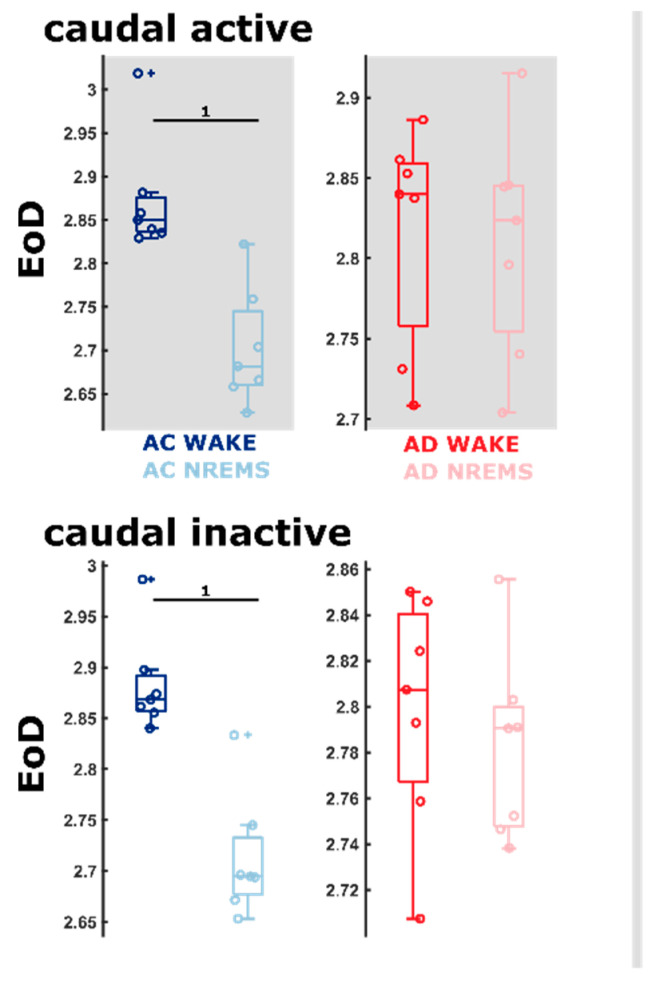
Comparison of entropy of difference (EoD) between WAKE and NREM sleep (NREMS). Combined box and dot plots of the EoD (τ = 1), the comparison within groups between WAKE and NREM sleep. Blue indicates the control (AC) and red—the Alzheimer’s (AD) group. We observed significant EoD differences (as indicated by AUC) in the AC group, but not in the AD group. The “+” indicate outlier for the box plot.

**Figure 7 ijms-21-09290-f007:**
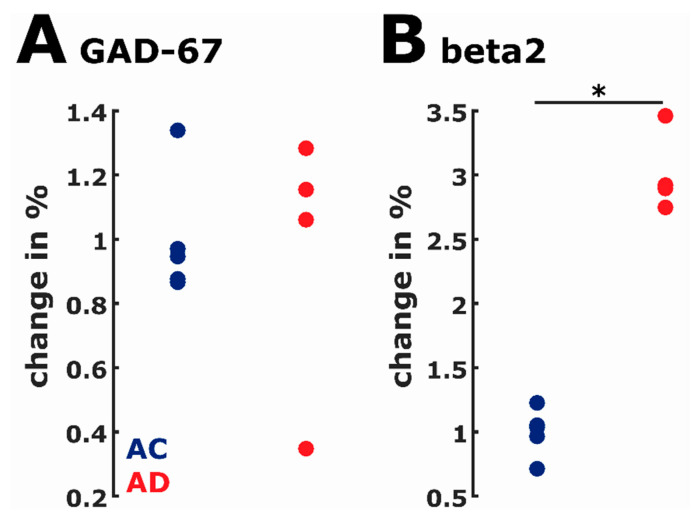
Dot plots of the relative change in protein expression normalized to beta-actin expression. (**A**) GAD-67 expression was not different between the groups indicating no significant change (AUC: 0.60, 95% CI: 0.2 to 1) in the amount of GABA produced; note that the highest value in the AC group and the lowest value in the AD group could be considered outliers as evaluated by the Grubbs’s test. (**B**) GABA_A_R β2 expression was significantly (AUC = 1) higher in the AD group. This indicates an increased number of GABA_A_R in the AD group. * indicates significance

**Figure 8 ijms-21-09290-f008:**
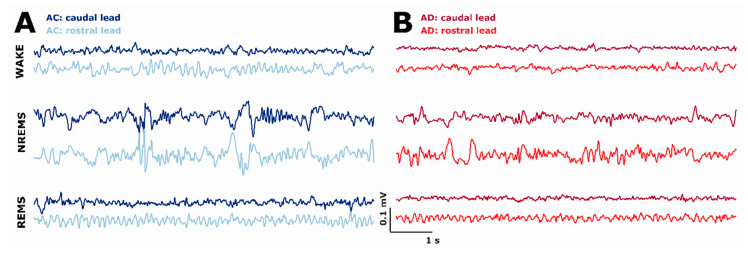
Exemplary EEG raw traces for the animals from (**A**) the age-matched controls and (**B**) the AD rats at different vigilance states—WAKE, NREM sleep, and REM sleep—that were recorded from the caudal or rostral EEG lead. NREMS: NREM sleep; REMS: REM sleep.

**Table 1 ijms-21-09290-t001:** Results of the AUC analysis to determine differences in the EoD (τ = 1) between the AC group and the AD group at certain vigilance states, as well as between WAKE and NREM sleep (NREMS) in either the AC or AD animals. Bold values indicate significance.

	**AC vs. AD**	**WAKE**	**NREMS**	**REMS**
Inactive	**Caudal**	0.59 [0.27–0.88]	**0.12 [0–0.35]**	**0.75 [0.44–1]**
Active	**Caudal**	**0.96 [0.81–1]**	**0.14 [0–0.43]**	**0.86 [0.57–1]**
	**WAKE vs. NREMS**	**AC**	**AD**	
Inactive	**Caudal**	**1**	0.57 [0.24–0.88]	
Active	**Caudal**	**1**	0.65 [0.33–0.96]

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
