# Peer review of "Sleep/Wake Behavior and EEG Signatures of the TgF344-AD Rat Model at the Prodromal Stage"

_ijms, 2020, doi:10.3390/ijms21239290_

Round 1
Reviewer 1 Report
Thank you for the opportunity to review this intriguing manuscript detailing sleep/wake and EEG markers of impending Alzheimer's disease in a rodent model. The authors present an examination of behavioral (sleep/wake), EEG, and protein (GABA-related) biomarkers of prodromal Alzheimer's Disease in an established rat model of AD. This study has a number of strengths, including the use of a rodent model that better recapitulates pathologic processes present in human disease, focus on markers during the prodromal phase of disease (when irreversible damage may not have occurred,) and the use of readily translatable markers.
The most compelling portion of this work is the evident fragmentation in sleep/wake exhibited in this rodent disease model. My only qualm with this portion of the investigation is the inconsistent use of epoch between the behavioral and EEG portion of the study. I recognize that this was done for expediency's sake, both 4s and 10s epochs have been proven as valid for analysis, and both are similarly extensively used in the literature. It has been suggested, however, that while the two epoch-lengths are equally valid for classification purposes, for determination of number of transitions, 4s epochs may be superior (Yan et al. 2011.) As such, I think it behooves the authors to examine both epoch lengths to fully evaluate differences in sleep/wake architecture.
The authors have chosen relevant leads for EEG analysis, as prefrontal and hippocampal changes have both been demonstrated in AD. Where significant qEEG differences between AD and control mice occur, however, does not seem intuitive, and I would appreciate some additional justification, perhaps in the discussion, as to why rostral leads were emphasized for spectral comparisons and caudal leads for information theoretic measures. I do recognize that the authors analyzed both, and that the caudal spectral anaylsis and rostral entropic measures are included in the supplemental material, but I believe that additional discussion presenting a hypothesis-driven testing schema or justification of differentially presented qEEG analyses would do much to improve the manuscript.
Finally, it is not entirely clear why you measured a GABAergic neuronal marker and a common GABA receptor subunit. While abstractly interesting, and relevant to the lab's interests, the relevance to the story presented was not made abundantly clear. In the discussion, you briefly mention changes in glutatmatergic signaling and possible imbalances in excitatory and inhibitory signaling in AD as something worth exploring. Why not then look at a combination of glutamatergic expression markers and gabaergic expression markers to look for that specific imbalance if that’s your hypothesis?
Minor Points:
- Lines 93, 154, 194, 377: Some sort of macro error apparently occurred
- Figure 6: Not entirely clear that wake is darker and NREM lighter, but that’s what I surmised from text.
- Figure 7: What is baseline here, beta-actin expression? You state in methods that you are normalizing to beta-actin expression, but it is not clear from the legend text what sample is the baseline to which you are comparing.
Thank you again for the opportunity to review your manuscript and I look forward to your responses.
Reference
Yan MM, Xu XH, Huang ZL, Yao MH, Urade Y, Qu WM. Selection of optimal epoch duration in assessment of rodent sleep–wake profiles. (2011) Sleep and Biological Rhythms;9: 46–55.
Author Response
Reviewer 1:
Thank you for the opportunity to review this intriguing manuscript detailing sleep/wake and EEG markers of impending Alzheimer's disease in a rodent model. The authors present an examination of behavioral (sleep/wake), EEG, and protein (GABA-related) biomarkers of prodromal Alzheimer's Disease in an established rat model of AD. This study has a number of strengths, including the use of a rodent model that better recapitulates pathologic processes present in human disease, focus on markers during the prodromal phase of disease (when irreversible damage may not have occurred,) and the use of readily translatable markers.
The most compelling portion of this work is the evident fragmentation in sleep/wake exhibited in this rodent disease model. My only qualm with this portion of the investigation is the inconsistent use of epoch between the behavioral and EEG portion of the study. I recognize that this was done for expediency's sake, both 4s and 10s epochs have been proven as valid for analysis, and both are similarly extensively used in the literature. It has been suggested, however, that while the two epoch-lengths are equally valid for classification purposes, for determination of number of transitions, 4s epochs may be superior (Yan et al. 2011.) As such, I think it behooves the authors to examine both epoch lengths to fully evaluate differences in sleep/wake architecture.
Thanks for the compliments on our work and for this thoughtful comment. We agree that the sleep fragmentation in the AD animal is an interesting finding. A possible reason for the increased fragmentation may be the changes in the EEG that seem to indicate a “more wake-like” EEG in the AD animals during sleep which (conceptually) appears to lower the threshold to transition between wake and sleep. Your comment regarding the transitions is a good one and we added information regarding the transition frequency for the 4 s scoring in this revision. See the added information in Results and to Figure 4.
Further we added the following text to the discussion
All results from quantitative EEG analysis were obtained from a distinct 2 h recording period (refer to the methods section for details), scored in 4 s EEG episodes rather than 10 s for a higher resolution. Epoch length of 4 s or 10 s are broadly used in rodent sleep research, although when dealing with analysis of stage transitions for example, a 4 s epoch has explicitly been suggested (Yan, Xu et al. 2011). This is in line with our experience in sleep scoring applying 4 s epochs to score fast and abrupt transitions (Fulda, Romanowski et al. 2011, Kreuzer, Polta et al. 2015)
The authors have chosen relevant leads for EEG analysis, as prefrontal and hippocampal changes have both been demonstrated in AD. Where significant qEEG differences between AD and control mice occur, however, does not seem intuitive, and I would appreciate some additional justification, perhaps in the discussion, as to why rostral leads were emphasized for spectral comparisons and caudal leads for information theoretic measures. I do recognize that the authors analyzed both, and that the caudal spectral anaylsis and rostral entropic measures are included in the supplemental material, but I believe that additional discussion presenting a hypothesis-driven testing schema or justification of differentially presented qEEG analyses would do much to improve the manuscript.
Thank you for this comment. Earlier versions of the manuscript presented all electrode data – but our feeling was that with that much data presented some of the themes were lost. We justify our reasoning thusly in the Discussion of this revised manuscript.
It should be noted that rodents have, by comparison to humans, a thinner neocortex relative to brain size. Therefore, oscillations from deeper structures make up a larger proportion of the overall EEG power with cranial electrodes. This is especially relevant for theta power, best seen over the hippocampal formation. In the rodent, theta power in our rostral electrode is strongly indicative of the wake and REM states (see Figure 5). Cortiocortical communication was the focus in our examination of complexity (see Figure 6). For these results we elected to emphasize the caudal lead (over the prefrontal cortex) as changes in the information content in the rostral lead might be assumed to contain more information from the hippocampus.
Finally, it is not entirely clear why you measured a GABAergic neuronal marker and a common GABA receptor subunit. While abstractly interesting, and relevant to the lab's interests, the relevance to the story presented was not made abundantly clear. In the discussion, you briefly mention changes in glutatmatergic signaling and possible imbalances in excitatory and inhibitory signaling in AD as something worth exploring. Why not then look at a combination of glutamatergic expression markers and gabaergic expression markers to look for that specific imbalance if that’s your hypothesis?
Thank you for allowing us to expand on this interesting finding. We rewrote several paragraphs in the Discussion based on your points and those of Reviewer 2.
It is known that this rodent model exhibits signs of increased cortical excitability. Interestingly, treatment with a drug that increases hippocampal signaling (cholinergic) appears to mitigate this seizure-like activity (Stoiljkovic, Kelley et al. 2018). The failure of the rats in our AD group to stabilize their quiescent activity (sleeping) without intrusion of wake supports the notion that suppression of neuronal activity is inadequately achieved in this disease model while quiescent. Similarly, the intrusion of sleep during waking behaviors suggests instability among vigilance states rather than a simple bias towards cortical excitation. Given the previously published presence of Aβ and neurofibrillary tangles at 16 months in the TgF344-AD rat (Cohen, Rezai-Zadeh et al. 2013), we analyzed protein expression related to GABA transmission in order to probe for possible evidence of inhibitory networks alterations in the early disease state. Our molecular data suggests that maintaining appropriate cortical excitation in each vigilance state is complex in these animals. Although it appears that both AC and AD groups seem to make a similar amount of inhibitory neurotransmitter at the synapses (See Figure 7) the production of GABAA receptors appears to be upregulated in the disease model (increase in expression). These molecular results can only be considered preliminary evidence that changes in the synaptic balance might underlie the physiologic changes and several caveats must be considered.
First, molecular data in this study was only performed in a small number of animals. The protein expression analysis revealed significantly increased GABAAR β2 expression in the AD animals but no significant change in GAD-67, i.e., in inhibitory synaptic transmission. Because two outliers may cause the observation of no significant result, we can draw no definitive conclusion regarding a possible effect. This increased protein production may not translate to increased functional GABAA receptors on the neurons’ surface, likely resulting in altered inhibitory network function. However, we are aware of our limited sample size and much more study focused on disease-related changes in excitatory and inhibitory networks in this model is needed.
Second, although no exact age range has been established that marks the end of “early” Alzheimer’s disease in this model (Cohen, Rezai-Zadeh et al. 2013), it is important to consider that our molecular data comes from animals older than our physiologic data. Therefore, these data may represent a greater progression of disease severity. Lastly, we did not quantify all the potential proteins that could contribute to excitatory and inhibitory balance. Future studies focused on membrane expression, the influence of glutamatergic, cholinergic signaling and the proportion of extrasynaptic to intrasynaptic GABAA receptors will be necessary to form a more complete picture. Although the β2 subunit of the GABAA receptor might be considered a marker of general GABAA receptor levels (Speigel, Bichler et al. 2017) it is possible that expression of other GABA and non-GABA related proteins are responsible for these effects.
Minor Points:
- Lines 93, 154, 194, 377: Some sort of macro error apparently occurred
Thanks for spotting these flaws. These were issues with the figure referencing. We corrected them.
- Figure 6: Not entirely clear that wake is darker and NREM lighter, but that’s what I surmised from text.
Thanks! We apologize for the missing legend. We added the missing information to the plot.
- Figure 7: What is baseline here, beta-actin expression? You state in methods that you are normalizing to beta-actin expression, but it is not clear from the legend text what sample is the baseline to which you are comparing.
Yes, actin expression. We amended the Figure legend. Thanks!
Thank you again for the opportunity to review your manuscript and I look forward to your responses.
REFERENCES
Cohen, R. M., K. Rezai-Zadeh, T. M. Weitz, A. Rentsendorj, D. Gate, I. Spivak, Y. Bholat, V. Vasilevko, C. G. Glabe and J. J. Breunig (2013). "A transgenic Alzheimer rat with plaques, tau pathology, behavioral impairment, oligomeric aβ, and frank neuronal loss." The Journal of neuroscience 33(15): 6245-6256.
Fulda, S., C. P. Romanowski, A. Becker, T. C. Wetter, M. Kimura and T. Fenzl (2011). "Rapid eye movements during sleep in mice: high trait-like stability qualifies rapid eye movement density for characterization of phenotypic variation in sleep patterns of rodents." BMC neuroscience 12(1): 110.
Kreuzer, M., S. Polta, J. Gapp, C. Schuler, E. Kochs and T. Fenzl (2015). "Sleep scoring made easy—Semi-automated sleep analysis software and manual rescoring tools for basic sleep research in mice." MethodsX 2: 232-240.
Speigel, I., E. K. Bichler and P. S. Garcia (2017). "The Influence of Regional Distribution and Pharmacologic Specificity of GABAAR Subtype Expression on Anesthesia and Emergence." Front Syst Neurosci 11: 58.
Stoiljkovic, M., C. Kelley, T. L. Horvath and M. Hajós (2018). "Neurophysiological signals as predictive translational biomarkers for Alzheimer's disease treatment: effects of donepezil on neuronal network oscillations in TgF344-AD rats." Alzheimers Res Ther 10(1): 105.
Yan, M.-M., X.-H. Xu, Z.-L. Huang, M.-H. Yao, Y. Urade and W.-M. Qu (2011). "Selection of optimal epoch duration in assessment of rodent sleep-wake profiles." Sleep and Biological Rhythms 9(1): 46-55.
Reviewer 2 Report
The manuscript submitted by Kreuzer and coworkers: “Sleep/Wake Behavior and EEG signatures of the TgF344-AD rat model at the prodromal stage” deals with the impact of a transgenic mutation of two common genes related to familial Alzheimer’s Disease in rats on sleep behavior and various EEG features. The authors raised the question whether changes in these two parameters occurred in the TgF344-AD rat model at a prodromal stage and compared these changes to alterations observed in humans suffering from Alzheimer’s Disease. They discovered that TgF344-AD rats had more sleep-wake transitions and an increased probability of shorter REM and NREM bouts compared to control. Furthermore, TgF344-AD rats demonstrated less EEG information during wake states, but more information during NREM states. The authors concluded that the changes in EEG activity could be caused by an Alzheimer’s Disease-induced change in inhibitory network function.
The topic of characterizing animal models for Alzheimer’s Disease is a contemporary issue, the manuscript is well written, the methods are sound and the results are reliable. Hence, I strongly recommend the manuscript to be published in IJMS. There are only a few points I would like to raise:
1.) My first point concerns the protein expression results. The authors aim at investigating sleep/wake behavior and EEG signatures of the TgF344-AD rat model at a prodromal stage of Alzheimer’s Disease. Therefore, they used TgF344-AD rats at an age of 17 months. Different from this, the protein expression experiments were performed in rats at a median age of 18 months (AC group) and 20 months (AD group). In my opinion it is less interesting that the mild age differences were not considered to be statistically different but more important, whether TgF344-AD rats at an age of 20 months (range 17-23 months) still represent the prodromal stage of Alzheimer’s Disease. Thus, I suggest to provide additional information about the development of the disease in the TgF344-AD rat model.
2.) My second question is why GABAA receptors incorporating a β2 subunit were studied. It would help if the authors gave additional information regarding their decision for GABAAR β2.
3.) The results for the expression of GAD-67 are not convincing. A total of only 8 animals were tested and there is one clear outlier in each group which could have changed the result. I would like to encourage the authors to discuss this topic more cautiously.
Minor Points
1.) The caption on page 10, line 202, must read: "...in the amount of GABA.” Instead of:”…in the amount of GABAA.”
2.) The term: “Error! Reference source not found.” got lost in the following places: page 4, line 93; page 10, line 197; page 16, line 377
Well done, Christian
Author Response
Reviewer 2
Comments and Suggestions for Authors
The manuscript submitted by Kreuzer and coworkers: “Sleep/Wake Behavior and EEG signatures of the TgF344-AD rat model at the prodromal stage” deals with the impact of a transgenic mutation of two common genes related to familial Alzheimer’s Disease in rats on sleep behavior and various EEG features. The authors raised the question whether changes in these two parameters occurred in the TgF344-AD rat model at a prodromal stage and compared these changes to alterations observed in humans suffering from Alzheimer’s Disease. They discovered that TgF344-AD rats had more sleep-wake transitions and an increased probability of shorter REM and NREM bouts compared to control. Furthermore, TgF344-AD rats demonstrated less EEG information during wake states, but more information during NREM states. The authors concluded that the changes in EEG activity could be caused by an Alzheimer’s Disease-induced change in inhibitory network function.
The topic of characterizing animal models for Alzheimer’s Disease is a contemporary issue, the manuscript is well written, the methods are sound and the results are reliable. Hence, I strongly recommend the manuscript to be published in IJMS. There are only a few points I would like to raise:
1.) My first point concerns the protein expression results. The authors aim at investigating sleep/wake behavior and EEG signatures of the TgF344-AD rat model at a prodromal stage of Alzheimer’s Disease. Therefore, they used TgF344-AD rats at an age of 17 months. Different from this, the protein expression experiments were performed in rats at a median age of 18 months (AC group) and 20 months (AD group). In my opinion it is less interesting that the mild age differences were not considered to be statistically different but more important, whether TgF344-AD rats at an age of 20 months (range 17-23 months) still represent the prodromal stage of Alzheimer’s Disease. Thus, I suggest to provide additional information about the development of the disease in the TgF344-AD rat model.
The transition from “early” Alzheimer’s Disease” to “frank” disease is difficult to delineate (in both humans and pre-clinical models). Because the TgF344-AD model is so new and because the cost of keeping these animals for 2 years or more is so expensive little information has been published on this important point. The original paper describing the model reports that neurofibrillary tangles can be seen at 16 months and uses 24-month old animals to demonstrate “frank” Alzheimer’s pathology. We have amended the discussion to consider our results with this important caveat. See response to third comment.
2.) My second question is why GABAA receptors incorporating a β2 subunit were studied. It would help if the authors gave additional information regarding their decision for GABAAR β2.
The Beta-2 subunit of the GABAA receptor can be found in intrasynaptic, extrasynaptic, benzodiazepine sensitive and benzodiazepine insensitive GABA receptors. It is also known to possess binding sites for protein-protein interactions that mediate surface expression. We chose this subunit as a “generic” biomarker of GABAA receptors in general. It is true that a more detailed molecular analysis could yield different results (see response to next comment).
3.) The results for the expression of GAD-67 are not convincing. A total of only 8 animals were tested and there is one clear outlier in each group which could have changed the result. I would like to encourage the authors to discuss this topic more cautiously.
Thanks for the comment. You are correct with your assumption. We performed the Grubb’s test for outliers and the high value in the AC and the low value in the AD group were classified as those. Without the outliers, we would see an effect. We amended the legend of the figure in question:
Note that the highest value in the AC group and the lowest value in the AD group could be considered outliers as evaluated by the Grubbs's test.
We also integrated these three important comments with those of Reviewer 1 and rewrote several paragraphs in the Discussion.
It is known that this rodent model exhibits signs of increased cortical excitability. Interestingly, treatment with a drug that increases hippocampal signaling (cholinergic) appears to mitigate this seizure-like activity (Stoiljkovic, Kelley et al. 2018). The failure of the rats in our AD group to stabilize their quiescent activity (sleeping) without intrusion of wake supports the notion that suppression of neuronal activity is inadequately achieved in this disease model while quiescent. Similarly, the intrusion of sleep during waking behaviors suggests instability among vigilance states rather than a simple bias towards cortical excitation. Given the previously published presence of Aβ and neurofibrillary tangles at 16 months in the TgF344-AD rat (Cohen, Rezai-Zadeh et al. 2013), we analyzed protein expression related to GABA transmission in order to probe for possible evidence of inhibitory networks alterations in the early disease state. Our molecular data suggests that maintaining appropriate cortical excitation in each vigilance state is complex in these animals. Although it appears that both AC and AD groups seem to make a similar amount of inhibitory neurotransmitter at the synapses (See Figure 7) the production of GABAA receptors appears to be upregulated in the disease model (increase in expression). These molecular results can only be considered preliminary evidence that changes in the synaptic balance might underlie the physiologic changes and several caveats must be considered.
First, molecular data in this study was only performed in a small number of animals. The protein expression analysis revealed significantly increased GABAAR β2 expression in the AD animals but no significant change in GAD-67, i.e., in inhibitory synaptic transmission. Because two outliers may cause the observation of no significant result, we can draw no definitive conclusion regarding a possible effect. This increased protein production may not translate to increased functional GABAA receptors on the neurons’ surface, likely resulting in altered inhibitory network function. However, we are aware of our limited sample size and much more study focused on disease-related changes in excitatory and inhibitory networks in this model is needed.
Second, although no exact age range has been established that marks the end of “early” Alzheimer’s disease in this model (Cohen, Rezai-Zadeh et al. 2013), it is important to consider that our molecular data comes from animals older than our physiologic data. Therefore, these data may represent a greater progression of disease severity. Lastly, we did not quantify all the potential proteins that could contribute to excitatory and inhibitory balance. Future studies focused on membrane expression, the influence of glutamatergic, cholinergic signaling and the proportion of extrasynaptic to intrasynaptic GABAA receptors will be necessary to form a more complete picture. Although the β2 subunit of the GABAA receptor might be considered a marker of general GABAA receptor levels (Speigel, Bichler et al. 2017) it is possible that expression of other GABA and non-GABA related proteins are responsible for these effects.
Minor Points
- The caption on page 10, line 202, must read: "...in the amount of GABA.” Instead of:”…in the amount of GABAA.”
Thanks for spotting. We edited the caption accordingly
- The term: “Error! Reference source not found.” got lost in the following places: page 4, line 93; page 10, line 197; page 16, line 377
Thanks! We corrected these issues
Well done, Christian
References
Cohen, R. M., K. Rezai-Zadeh, T. M. Weitz, A. Rentsendorj, D. Gate, I. Spivak, Y. Bholat, V. Vasilevko, C. G. Glabe and J. J. Breunig (2013). "A transgenic Alzheimer rat with plaques, tau pathology, behavioral impairment, oligomeric aβ, and frank neuronal loss." The Journal of neuroscience 33(15): 6245-6256.
Speigel, I., E. K. Bichler and P. S. Garcia (2017). "The Influence of Regional Distribution and Pharmacologic Specificity of GABAAR Subtype Expression on Anesthesia and Emergence." Front Syst Neurosci 11: 58.
Stoiljkovic, M., C. Kelley, T. L. Horvath and M. Hajós (2018). "Neurophysiological signals as predictive translational biomarkers for Alzheimer's disease treatment: effects of donepezil on neuronal network oscillations in TgF344-AD rats." Alzheimers Res Ther 10(1): 105.
Round 2
Reviewer 1 Report
Thank you for addressing my concerns, I have no additional recommendations.